# Exploring the Relationship of Leisure Travel with Loneliness, Depression, and Cognitive Function in Older Adults

**DOI:** 10.3390/ijerph21040498

**Published:** 2024-04-18

**Authors:** Shu Cole, Chenggang Hua, Siyun Peng, Weixuan Wang

**Affiliations:** 1Department of Health and Wellness Design, Indiana University School of Public Health—Bloomington, Bloomington, IN 47405, USA; chhua@iu.edu (C.H.); wangweix@indiana.edu (W.W.); 2Department of Sociology, Indiana University—Bloomington, Bloomington, IN 47405, USA; siypeng@iu.edu

**Keywords:** aging, leisure travel, loneliness, depression, cognition, Health and Retirement Study

## Abstract

Loneliness, depression, and cognitive decline are pressing concerns among older adults. This study examines the association between leisure travel participation and these health outcomes in older adults, aiming to provide further evidence of the benefits of leisure travel. Using nationally representative historical data from the 2006 household survey of the Health and Retirement Study, this study conducted a series of regression analyses to investigate the relationship between traveling and the three health outcomes, adjusting for age, sex, race, marital status, education, total wealth, annual income, and difficulty with daily activities. The results reveal that travel patterns in terms of distance are significantly associated with loneliness, depression, and cognitive function. Long-distance travel is positively related to higher cognitive function and a reduction in depressive symptoms, along with lower levels of loneliness, reinforcing the notion that leisure travel can potentially act as a catalyst for improved cognitive and mental health by offering opportunities for enhancing social connections and forming new relationships. The findings on the relationships between participation in leisure travel and mental and cognitive health contribute to the body of evidence supporting the therapeutic value of leisure travel in promoting healthy aging and enhancing the overall well-being in older adults.

## 1. Introduction

As individuals age, they may experience declines in physical and cognitive functioning. Loneliness, depression, and dementia become prominent concerns for older adults, as these conditions are not typical aspects of the aging process. Recent research shows that over 40% of those aged 60 and older report feeling lonely [1]. Social isolation and loneliness have been found to strongly link to the development and exacerbation of depression, which, in turn, can contribute to cognitive decline and dementia [1,2,3,4]. Currently, Alzheimer’s Disease and Alzheimer’s Disease Related Dementias (AD/ADRD) rank as the fifth leading cause of death among adults aged 65 years and older [5]. The prevalence of AD/ADRD and the lack of a cure have prompted extensive research into identifying factors associated with AD/ADRD in an attempt to prevent and intervene in the disease. Research on factors that can promote social connectedness, alleviate loneliness, and delay cognitive decline is greatly needed.

Leisure activity, especially when undertaken out-of-home, may act as a mediator between social isolation and depressive symptoms, with out-of-home activities showing greater benefits for mental health [6]. Leisure travel, as an activity in the leisure and recreational dimension of life, has long been recognized as an effective means of socialization. Therefore, it has the potential to catalyze social connectedness, preventing loneliness, and delaying cognitive decline among the vulnerable population of older adults [7,8,9]. However, the clinically meaningful outcomes of leisure travel have not been fully established. The purpose of this study is to explore the association of leisure travel with loneliness, depression, and cognitive function among older adults, using a nationally representative sample from the historical 2006 Health and Retirement Study (HRS).

## 2. Literature Review

Loneliness and social isolation among individuals aged 65 and older are associated with depression, anxiety [10,11,12,13], and cognitive decline [14]. Social disconnection and a lack of meaningful social interactions can lead to loneliness, which, over time, can contribute to the development of depressive symptoms [15,16]. Depression, characterized by persistent sadness, loss of interest, and impaired cognitive functioning, is a significant risk factor for AD/ADRD [17,18,19]. Depressive symptoms before the onset of AD/ADRD are one of the early signs of the disease and can accelerate cognitive decline in those with mild cognitive impairment [20,21]. Depression’s negative impacts on the brain, including alterations in neuroplasticity, neuroinflammation, and neurodegeneration, can further contribute to cognitive decline and the manifestation of dementia symptoms [20].

A substantial proportion of individuals with depression frequently report persistent feelings of loneliness [22] and they are significantly more prone to experiencing loneliness than the general population [23]. Loneliness has been shown to predict poor recovery from anxiety and depression [24]. In a study by Coyle and Duganusing [25] using data from the 2006 and 2008 waves of the HRS (N = 11,825), they examined the association between social isolation, loneliness, and the likelihood of having a mental health problem. They found that both loneliness and objective social isolation were significantly related to the presence of a mental health problem.

Leisure activities are often recommended by senior care providers, as these activities can help improve and maintain older adults’ social networks [26]. The structure and characteristics of one’s social network, such as diverse and friend-centered networks, are associated with a lower prevalence of depressive symptoms [27]. Active leisure activities, such as sports, trips, museum visits, exhibitions, as well as dining out and parties, are particularly beneficial [11,28]. Leisure travel, whether overnight or day trips, is a multi-modal activity that engages travelers cognitively, physically, and psychosocially. It can stimulate older adults’ interest in life and enhance their social engagement, thus playing a vital role in promoting social connectedness and mitigating the negative impact of loneliness and social isolation on mental and physical health [29,30].

The benefits of travel have been widely researched in tourism studies [31,32], and to some extent in health science studies [33,34], with evidence indicating improved overall self-reported health [35], stress relief [36], mental and psychological well-being [37], and overall life satisfaction and quality of life [38,39]. Studies have consistently shown that leisure travel, in particular, offers significant benefits to older adults, including increased feelings of relaxation and well-being, as well as improvements in social relationships [40,41]. There is emerging clinical evidence demonstrating the positive effects of leisure travel on cognitive and mental health. For example, Pagan’s [42] analysis of a large panel of data with 17,624 respondents found that an increase in the intensity of participation in day trips or short trips significantly reduced respondents’ loneliness scores. Even taking short trips within one’s community can help prevent social isolation and loneliness experienced by many older adults [43,44].

There appears to be agreement among travel and tourism researchers that regular travel participation benefits older adults by helping them maintain their physical and mental health, promoting longevity, and contributing to successful aging [45,46,47]. Nevertheless, additional clinical evidence is needed to confirm the effects of leisure travel on the mental and cognitive health of older adults. As a preliminary step, this study aims to investigate the association of leisure travel with loneliness, depressive symptoms, and cognitive function in older adults. This study is expected to potentially add to the literature on clinically meaningful health outcomes of leisure travel. Subsequently, a better understanding of travel patterns and behavior and related health outcomes can assist health professionals in developing social engagement activities for older adults. Furthermore, the literature highlights that travel’s potential to enhance health and well-being frequently serves as a primary motivator for older adults to engage in travel and tourism activities [46,48]. This study’s results are expected to offer insights into travel businesses, encouraging them to enhance their capacity to provide travel services that optimize their impact on the social connectedness, mental well-being, and cognitive health of older adults.

## 3. Methods

### 3.1. Study Population

All analyses were conducted using data from the HRS, a biennial, nationally representative household survey conducted by the Institute for Social Research at the University of Michigan and sponsored by the National Institute on Aging. Currently, there is no nationally representative data known to have been collected specifically for studying leisure travel and the specified health outcomes. For this study, we extracted relevant variables utilizing data from the 2006 HRS survey, which is part of the longitudinal dataset harmonized by the RAND Center for the Study of Aging. The 2006 survey included a unique set of questions about specific types of travel activities in the past 12 months. Results from the data should provide us with preliminary findings on leisure travel and health outcomes from a historical perspective.

Figure 1 illustrates the data selection process. Among the HRS respondents in the 2006 wave, a total of 7730 participants completed the psychosocial survey, which included travel-related questions. Since the HRS targets respondents over 50 years old, our study only included respondents aged 51 years and older, resulting in an eligible sample size of 7457 respondents. The final primary analytic sample size is 7037 after excluding missing data in key outcome variables. The HRS provides sampling weights to adjust for non-response bias, ensuring that our estimations are representative of the population.

### 3.2. Measurements

Travel participation. The key independent variable in this study is travel participation, indicated by the range from local to international travel. The distance traveled signifies the levels of complexity in the activity and the cognitive, physical, and mental abilities required during the engagement process. The survey includes questions about whether respondents have taken specific types of trips in the last 12 months, including (1) vacations within the US, (2) vacations outside the US, and (3) day trips or outings. We categorized respondents into four groups: those who have not taken any trips (non-travelers), those who have taken only day trip/outings (local travelers), those who have taken domestic trips (domestic travelers), and those who have taken international trips (international travelers). This study designates the four groups on a continuum of distances traveled, with non-travelers and international travelers representing the two extremes. International travel is considered the longest distance traveled, followed by domestic and local trips.

Cognitive functioning. Cognition was assessed using a battery of cognitive tests: (1) immediate and delayed recall of 10 nouns (0 to 20 points), (2) a serial of 5 subtractions by the number of 7 (0 to 5 points), and (3) a backward count task from 20 (0–2 points). We adopted the Langa-Weir Classifications [49] of cognitive functioning, which included the sum of all test scores. The total test scores ranged from 0 to 27 points, with higher scores indicating better cognitive functioning.

Depressive symptoms. Depression was measured using the modified Center for Epidemiologic Studies–Depression scale (CES-D) which included eight questions regarding whether the respondent (1) felt depressed; (2) thought everything was an effort; (3) slept restlessly; (4) felt lonely; (5) felt sad; (6) felt happy; (7) enjoyed life, and (8) could not get going [50]. Items 6 and 7 were inversely coded, and the total number of “yes” responses are summed as the CES-D score (0 to 8), with higher numbers indicating more depressive mood.

Loneliness. Loneliness was measured with the short, modified UCLA Loneliness Scale [51]. Respondents were asked how frequently (often, some of the time, hardly ever/never) they felt (1) lacking companionship, (2) left out, and (3) isolated. The total loneliness scores ranged from 0 to 9. We inversely coded the variables, with higher values indicating higher levels of loneliness.

Covariates. Potential confounders in this study were identified through a review of the literature. Sociodemographic characteristics included age, sex, race (White and all others), number of years in education, marital status (married or living with a partner, and other), total wealth (total household assets minus debt, including second home), and household total income of the last calendar year. To account for the wide variations in these variables, we categorized wealth into quartiles (≤USD 65,000; USD 65,001–233,500; USD 233,501–570,000; >USD 570,000) and income into quartiles (≤USD 20,051; USD 20,052–39,200; USD 39,201–74,775; >USD 74,775) in our analysis.

Previous studies have demonstrated that a person’s physical health is associated not only with cognitive health and depression [52] but also with travel participation [53]. Therefore, difficulties in daily activities, measured with the Instrumental Activities of Daily Living Scale (IADL), were included as a covariate. IADL was coded as a dummy variable, with 1 assigned if the respondent reported one or more of the following difficulties: (1) using the phone, (2) managing money, (3) taking medications, (4) shopping for groceries, and (5) preparing hot meals.

### 3.3. Statistical Analysis

We began with a descriptive analysis to assess whether the four groups exhibited differences in the outcome variables and confounding factors. For continuous variables, we calculated the means (M) and standard deviations (±SD), followed by group mean comparisons using one-way analysis of variance (ANOVA) tests (F). For categorical variables, we presented the frequencies as percentages (%), followed by Chi-squared (χ^2^) tests to evaluate group differences.

Next, we performed linear regression analyses to investigate the association between travel participation and the outcomes of cognitive function and mental health variables (i.e., loneliness and depression). Multivariable general linear models were adopted to calculate estimated mean scores for each outcome while controlling for potential confounders (age, sex, race, education, marital status, wealth, income, and IADL).

We employed listwise deletion to address missing data since imputations are not recommended for outcome variables [54]. We applied weights to all models to account for bias due to non-response and sampling design. Additionally, we conducted regression analyses without adjusting for sample weights to assess the robustness of our findings. Variables with *p*-values ≤ 0.05 in the current analyses were considered significant factors associated with the outcomes. Analyses were performed using Stata 17 [55].

## 4. Results

Table 1 presents population estimates for the four traveler groups (N_int’l_ = 1111; N_domestic_ = 1741; N_local_ = 3239; N_notrip_ = 946). Significant group differences were found in all outcomes and demographic characteristics (*p* < 0.001), except marital status. Compared to non-travelers, travelers were younger (M_int’l_ = 63.25 years old, SD_int’l_ = 8.72; M_domestic_ = 64.39, SD_domestic_ = 9.26; M_local_ = 68.19, SD_local_ = 10.24; M_notrip_ = 69.83, SD_notrip_ = 10.88) and had higher levels of wealth (W > 570,001_int’l_ = 43%; W > 570,001_domestic_ = 29%; W > 570,001_local_ = 16%; W > 570,001_notrip_ = 10%), annual income (I > 74,775_int’l_ = 51%; I > 74,775_domestic_ = 36%; I > 74,775_local_ = 17%; I > 74,775_notrip_ = 7%), and education (M_int’l_ = 14.12 years, SD_int’l_ = 3.17; M_domestic_ = 13.41, SD_domestic_ = 2.60; M_local_ = 12.29, SD_local_ = 2.86; M_notrip_ = 10.88, SD_notrip_ = 3.51). They were more likely to be White (Race_int’l_ = 89%; Race_domestic_ = 89%; Race_local_ = 85%; Race_notrip_ = 78%) and had fewer difficulties in daily living activities (IADL_int’l_ = 5%; IADL_domestic_ = 7%; IADL_local_ = 18%; IADL_notrip_ = 29%).

Significant differences were observed among the four groups in terms of loneliness, depressive symptoms, and cognitive function (all *p*s < 0.001). Longer-distance travelers reported better cognitive and mental health. They reported lower levels of loneliness (M_int’l_ = 1.36, SD_int’l_ = 0.47; M_domestic_ = 1.44, SD_domestic_ = 0.50; M_local_ = 1.59, SD_local_ = 0.54; M_notrip_ = 1.76, SD_notrip_ = 0.64), fewer depressive symptoms (M_int’l_ = 0.87, SD_int’l_ = 1.49; M_domestic_ = 1.10, SD_domestic_ = 1.65; M_local_ = 1.84, SD_local_ = 2.02; M_notrip_ = 2.53, SD_notrip_ = 2.37), and higher cognitive function (M_int’l_ = 17.32, SD_int’l_ = 3.98; M_domestic_ = 16.53, SD_domestic_ = 3.92; M_local_ = 14.75, SD_local_ = 4.43; M_notrip_ = 12.60, SD_notrip_ = 4.43).

To investigate the association of travel participation with cognitive function, depression, and loneliness, separate linear regression analyses were conducted for each outcome, while controlling for confounding variables. We compared baseline scores for total cognitive function, the total number of depressive symptoms, and loneliness across different types of trip participation, adjusting for all covariates in each linear regression. The results from the multivariable-adjusted models are summarized in Table 2.

Overall, local travelers, domestic travelers, and international travelers differed significantly from non-travelers in their cognitive function (*p* < 0.001), depressive symptoms (*p* < 0.001), and loneliness (*p* < 0.001), even after adjusting for age, sex, race, education, marital status, IADL, income, and wealth. Specifically, respondents who engaged in local trips (b = 0.79; *p* < 0.001), domestic trips (b = 1.14; *p* < 0.001), and international trips (b = 1.32; *p* < 0.001) reported higher cognitive function compared to those who did not travel in the last 12 months. Similarly, local, domestic, and international travelers reported lower levels of loneliness (b = −0.13, b = −0.24, b = −0.29, respectively; all *p*s < 0.001) and fewer depressive symptoms (b = −−0.31, b = −0.86, b = −0.94, respectively; all *p*s < 0.001) than non-travelers.

The results in Table 2 demonstrate a positive relationship between travel participation and cognitive function, as well as a negative relationship with loneliness and depressive symptoms. Figure 2 visually illustrates these associations between traveling and the three health outcomes. Notably, there is a clear dose-response relationship between travel distance and health outcomes, indicated by the slope in each plot. International travelers reported the best health across all three outcomes, with the highest cognitive function, the lowest level of loneliness, and the fewest depressive symptoms. They were followed by domestic travelers, then local travelers, and finally, non-travelers, who exhibited the lowest cognitive function, the highest level of loneliness, and the most depressive symptoms.

### Sensitivity Analysis

To assess the robustness of this study’s findings, we conducted regression analyses examining the associations between traveling and the three health outcomes using the sample data without applying weights. As shown in Table 3, the results of the regression analyses, adjusting for all covariates, exhibit similar patterns to those obtained using weighted sample data when comparing traveling groups to non-travelers as the reference group.

## 5. Discussion

Little research has been devoted to investigating the association between travel participation and loneliness, depression, and cognitive function. Previous efforts have been predominantly qualitative. This study is unique in its utilization of nationally representative samples to quantitatively examine the relationship of travel participation with mental and cognitive health outcomes. The results reveal significant associations between travel patterns and the three health outcomes. Travel participation was operationalized based on varying distances traveled, from not leaving home to local or day trips, to domestic and international trips, reflecting the complexity of different travel experiences. Long-distance travel typically involves less familiarity with the destination, requiring greater physical, mental, and cognitive engagement. As this study finds, long-distance travel is associated with higher cognitive function and fewer depressive symptoms. These findings align with previous qualitative studies [56,57], supporting their findings on the positive effects of long-distance travel on mental well-being and cognitive health. Our findings provide further empirical evidence reinforcing the notion that the extent of travel, particularly long-distance travel, is linked to better cognitive health and decreased depressive symptoms.

The negative relationship between leisure travel participation and loneliness may be attributed to the socialization benefits of travel, as evidenced in the travel and tourism literature. Existing research has demonstrated that engaging in leisure travel usually offers individuals opportunities to connect with others, form new social bonds, and strengthen existing relationships [58,59,60,61]. Travelers step beyond their immediate social circles, exploring new destinations and exposing themselves to diverse social environments, cultures, and communities, which fosters social interaction and engagement [36,62]. As suggested by previous research, these socialization opportunities help alleviate feelings of loneliness and isolation by enhancing a sense of belonging, connection, and shared experiences with others [30,63]. Furthermore, the positive social interactions and relationships formed during travel may generate a greater sense of social connectedness and thus enrich their well-being [64,65]. Although social interactions encountered during travel may not all be meaningful or long-term, they have the potential to reduce the loneliness of travelers. For instance, Song et al.’s study indicated that interactions with employees of businesses had a significant effect on reducing the loneliness of senior travelers [66].

The detrimental impact of loneliness on mental well-being and cognitive function has consistently been observed in the literature [12,13,67,68]. These findings underscore the need for targeted interventions and support systems to address loneliness, as mitigating this risk factor could potentially contribute to improved cognitive health and reduced risk of depression. There is increasing evidence in the literature that engaging in leisure activities may decrease the risk for AD/ADRD. Kuiper et al.’s systematic review and meta-analysis of 19 longitudinal cohort studies found an increased risk of dementia with low levels of social engagement, such as participating in leisure activities [69]. The negative relationship between leisure travel and loneliness found in this study reaffirms the potential significance of engagement in leisure activities.

Examining the association of travel with cognitive and mental health is both novel and significant. In our study, travel participation, whether local or long-distance, was associated with improved cognitive function and a reduced number of depressive symptoms and loneliness compared to non-travelers. Notably, long-distance travelers exhibited the most favorable outcomes, including the lowest prevalence of cognitive impairment, depression, and loneliness. These results suggest the importance of identifying patterns of travel behavior and how they relate to health outcomes. Studies designed to examine the impact of leisure travel on mental and cognitive health are rare, and research on the topic using nationally representative data is non-existent. Our study is innovative in using secondary data to provide quantitative evidence that lays the foundation for further research. Future research should delve into the underlying mechanisms through which travel connects with these outcomes and further explore its potential in promoting cognitive health and emotional well-being.

### 5.1. Limitations and Future Studies

Firstly, this study solely explores the association of leisure travel participation with mental and cognitive health outcomes among older adults, using self-reported cross-sectional data. Consequently, its results do not provide evidence for the impact of travel activities on cognition health or mental health. Additionally, we acknowledge the potential of “reverse causality,” where high levels of leisure travel participation may not influence mental and cognitive health; rather, high levels of mental and cognitive health may determine a person’s participation in travel activities. With the emerging clinical evidence of leisure activities on health outcomes, the relationship between leisure travel and health outcomes can be reciprocal. Comprehensive reviews of recent research findings on leisure activities and health outcomes are much needed. Well-designed experiments or longitudinal studies examining the temporal relationship between travel and health outcomes may provide insights into whether travel can influence cognitive and mental health.

Secondly, although it is valuable that this study utilized secondary data from a nationally representative sample to investigate the relationship between leisure travel and mental and cognitive health, the data were collected in 2006, which may not accurately reflect the travel behavior and health outcomes of current older adults. Therefore, there is a pressing need for up-to-date quantitative data on the salient variables from nationally representative samples.

Thirdly, while the regression analyses were adjusted for several confounding variables, the included variables were limited. For instance, research indicates that physical health conditions, such as vascular health, are risk factors for AD/ADRD. Therefore, future studies should aim to identify and adjust for more comprehensive confounders based on current clinical and research findings.

Fourthly, the study acknowledges that not everyone is interested in or able to engage in long-distance travel. Additionally, as individuals age, their participation in long-distance travel is expected to decline. Hence, the findings may not apply to all segments within the older adult population. Future studies should examine the value of travel among diverse groups of older adults.

Lastly, while the study focuses on the associations between travel and health outcomes, it does not quantify the strength of each type of travel participation to each health outcome. Future research may examine these relationships’ strengths to assess the magnitude of travel’s influence as a sub-dimension of an active lifestyle.

### 5.2. Implications

Despite the limitations of this study, its findings offer quantitative evidence of how leisure travel activities, such as local sight-seeing tours and holiday trips, are related to mental and cognitive health outcomes. This opens a promising horizon for further research focusing on the potential role of leisure travel in promoting mental and cognitive health.

A better understanding of the linkages between various travel behaviors and health outcomes can assist health professionals, including recreation therapists, in designing social engagement activities more effectively. For older adults who are experiencing depression or cognitive decline, they may not choose to engage in leisure travel. Nevertheless, encouraging individuals to participate in leisure travel may enhance social interactions and, consequently, improve their mental and cognitive health.

Public health initiatives could incorporate leisure outings, both local and long-distance, as part of broader efforts to promote cognitive health and mental well-being. Encouraging individuals to engage in travel experiences, even on a local scale, may have beneficial effects on cognitive function, mental health, and social integration. These initiatives could be especially valuable for vulnerable populations, including older adults or individuals at risk of cognitive decline or mental health issues. The significant associations between travel participation and cognitive function, as well as mental health outcomes, suggest that policymakers should focus on creating supportive environments that facilitate travel opportunities, improving accessibility for diverse populations, and integrating travel-related initiatives into healthcare systems.

## 6. Conclusions

In examining the potential association between leisure travel participation and mental and cognitive health outcomes among older adults, this study highlights the importance of social engagement, particularly through activities like leisure travel, in mitigating loneliness, depression, and cognitive decline. Leisure travel, despite the distance traveled, offers opportunities for individuals to connect with others, form new social bonds, and strengthen existing relationships. Further research employing robust methodologies is essential to further investigate the association between travel and mental health outcomes, as well as to explore the underlying mechanisms driving these associations. Backed by valid research evidence, public health initiatives could integrate travel-related activities into broader strategies aimed at promoting healthy aging and mitigating cognitive decline and mental health challenges.

## Figures and Tables

**Figure 1 ijerph-21-00498-f001:**
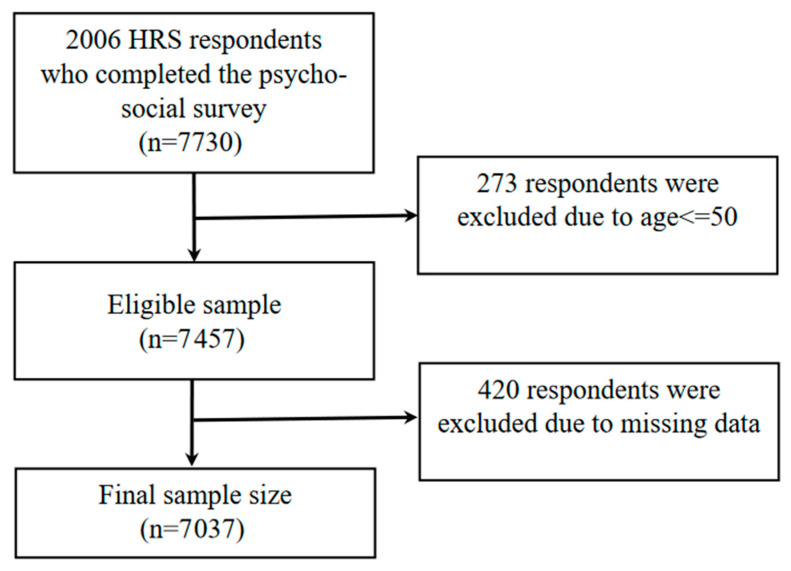
The flowchart illustrating the data selection process.

**Figure 2 ijerph-21-00498-f002:**
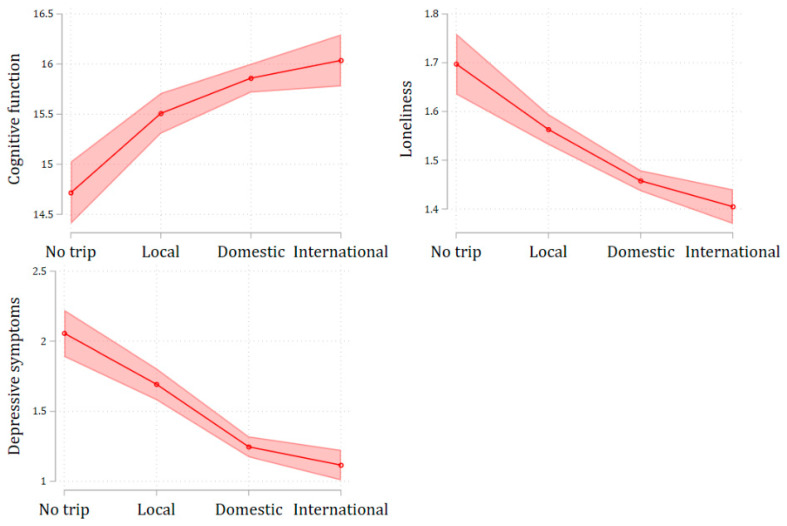
Plots of marginal effects of regression models on the relationships between traveling and health outcome.

**Table 1 ijerph-21-00498-t001:** Demographic and health characteristics of the study groups (N = 7037).

	No Trip (N = 1111)		Local (N = 1741)		Domestic (N = 3239)		International (N = 946)		F/χ^2^
	Pop. M (%)	SD	Pop. M (%)	SD	Pop. M (%)	SD	Pop. M (%)	SD	
White	(78)		(85)		(89)		(89)		*p <* 0.001
Women	(54)		(56)		(55)		(54)		*p* > 0.050
Married	(54)		(59)		(73)		(77)		*p <* 0.001
IADL	(29)		(18)		(07)		(05)		*p <* 0.001
Household Income									*p <* 0.001
*≤USD 20,051*	(49)		(31)		(14)		(10)		
*USD 20,052–39,200*	(26)		(29)		(20)		(16)		
*USD 39,201–74,775*	(18)		(23)		(29)		(23)		
*>USD 74,775*	(07)		(17)		(36)		(51)		
Household Wealth									*p <* 0.001
*≤USD 65,000*	(47)		(33)		(17)		(11)		
*USD 65,001–233,500*	(27)		(30)		(25)		(16)		
*USD 233,501–570,000*	(16)		(20)		(29)		(30)		
*>USD 570,000*	(10)		(16)		(29)		(43)		
Age	69.83	10.88	68.19	10.24	64.39	9.26	63.25	8.72	*p <* 0.001
Education	10.88	3.51	12.29	2.86	13.41	2.60	14.12	3.17	*p <* 0.001
Cognitive function	12.60	4.43	14.75	4.43	16.53	3.92	17.32	3.98	*p <* 0.001
Loneliness	1.76	0.64	1.59	0.54	1.44	.50	1.36	0.47	*p <* 0.001
Depressive symptoms	2.53	2.37	1.84	2.02	1.10	1.65	0.87	1.49	*p <* 0.001

Note: Pop. = Population; M = Mean; SD = Standard deviation; IADL = Instrumental activities of daily living.

**Table 2 ijerph-21-00498-t002:** Linear regression results on the relationship between traveling and health outcomes (adjusted for weights).

	Cognitive Function	Loneliness	Depressive Symptoms
* Travel (Ref. = No trip) *			
Local	0.792 ***	−0.134 ***	−0.363 ***
	(0.174)	(0.032)	(0.101)
Domestic	1.143 ***	−0.239 ***	−0.810 ***
	(0.182)	(0.034)	(0.087)
International	1.321 ***	−0.293 ***	−0.941 ***
	(0.214)	(0.037)	(0.102)
White	1.956 ***	0.024	−0.013
	(0.165)	(0.027)	(0.076)
Education	0.418 ***	−0.000	−0.065 ***
	(0.025)	(0.003)	(0.010)
Women	0.694 ***	−0.008	0.133 **
	(0.106)	(0.017)	(0.041)
Age	−0.121 ***	−0.008 ***	−0.023 ***
	(0.005)	(0.001)	(0.003)
Married	−0.315 *	−0.210 ***	−0.338 ***
	(0.126)	(0.021)	(0.057)
IADL	−1.479 ***	0.171 ***	1.415 ***
	(0.191)	(0.026)	(0.087)
* Household income (Ref. ≤ USD 20,051) *			
USD 20,052–39,200	0.706 ***	−0.021	−0.363 ***
	(0.194)	(0.028)	(0.091)
USD 39,201–74,775	0.800 ***	−0.030	−0.369 **
	(0.181)	(0.030)	(0.108)
>74,775	0.982 ***	−0.034	−0.441 ***
	(0.200)	(0.032)	(0.107)
* Household wealth (Ref. ≤ USD 65,000) *			
USD 65,001–233,500	0.526 ***	−0.056	−0.291 ***
	(0.120)	(0.032)	(0.075)
USD 233,501–570,000	0.561 ***	−0.105 ***	−0.094
	(0.123)	(0.028)	(0.087)
>USD 570,000	0.922 ***	−0.139 ***	−0.239 **
	(0.163)	(0.027)	(0.083)
N	7036	7036	7036
R-squared	0.385	0.126	0.195

* *p* < 0.05; ** *p* < 0.01; *** *p* < 0.001. Standard errors in parentheses. Ref. = Reference group.

**Table 3 ijerph-21-00498-t003:** Linear regression results on the relationship between traveling and health outcomes (without weights).

	Cognitive Function	Loneliness	Depressive Symptoms
* Travel (Ref. = No trip) *			
Local	0.702 ***	−0.131 ***	−0.431 ***
	(0.148)	(0.023)	(0.080)
Domestic	1.090 ***	−0.229 ***	−0.798 ***
	(0.141)	(0.022)	(0.076)
International	1.185 ***	−0.265 ***	−0.935 ***
	(0.171)	(0.026)	(0.088)
White	2.014 ***	0.035	0.062
	(0.121)	(0.018)	(0.063)
Education	0.411 ***	−0.003	−0.065 ***
	(0.017)	(0.002)	(0.009)
Women	0.692 ***	−0.000	0.159 ***
	(0.087)	(0.013)	(0.043)
Age	−0.121 ***	−0.007 ***	−0.021 ***
	(0.005)	(0.001)	(0.003)
Married	−0.307 **	−0.204 ***	−0.399 ***
	(0.109)	(0.017)	(0.058)
IADL	−1.328 ***	0.190 ***	1.331 ***
	(0.154)	(0.022)	(0.084)
* Household income (Ref. ≤ USD 20,051) *			
USD 20,052–39,200	0.703 ***	−0.020	−0.337 ***
	(0.137)	(0.020)	(0.070)
USD 39,201–74,775	0.791 ***	−0.021	−0.305 ***
	(0.152)	(0.022)	(0.077)
>USD 74,775	0.870 ***	−0.023	−0.330 ***
	(0.166)	(0.024)	(0.083)
* Household wealth (Ref. ≤ USD 65,000) *			
USD 65,001–233,500	0.574 ***	−0.064 **	−0.213 **
	(0.131)	(0.020)	(0.068)
USD 233,501–570,000	0.601 ***	−0.108 ***	−0.137
	(0.137)	(0.021)	(0.072)
>USD 570,000	0.897 ***	−0.131 ***	−0.223 **
	(0.149)	(0.022)	(0.075)
N	7037	7037	7037
R-squared	0.369	0.125	0.183

* *p* < 0.05; ** *p* < 0.01; *** *p* < 0.001. Standard errors in parentheses. Ref. = Reference group.

## Data Availability

Data can be found at https://hrsdata.isr.umich.edu/data-products, accessed on 11 June 2023.

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
