# Peer review of "Exploring the Relationship of Leisure Travel with Loneliness, Depression, and Cognitive Function in Older Adults"

_ijerph, 2024, doi:10.3390/ijerph21040498_

Round 1

Reviewer 1 Report (Previous Reviewer 1)

Comments and Suggestions for Authors

previous points have been explained. 

Author Response

Thank you for your support!

Reviewer 2 Report (Previous Reviewer 2)

Comments and Suggestions for Authors

Thank you for considering my comments. Now, the manuscript represents a better quality. It meets all the journal's requirements.

Comments on the Quality of English Language

Minor editing of English language required.

Author Response

We have edited the manuscript. Thank you!

This manuscript is a resubmission of an earlier submission. The following is a list of the peer review reports and author responses from that submission.

Round 1

Reviewer 1 Report

Comments and Suggestions for Authors

The paper deals with the correlation of travelling with loneliness depression and dementia at elderly. Although there is an interest in correlating age with travelling and other psychological situations, there is an impelling need to prove also causality in this relation (as mentioned in the closing part) and analyze whether the independent variables are indeed the dependent ones

Other points;

tables should be more accurately named and described

references are as superscript in the current version

novelty of the study should be highlighed compared with previous findings

Author Response

We appreciate the constructive comments provided by Reviewer 1. We are confident that the manuscript has been significantly improved following the revisions made in response to the reviewer’s valuable suggestions. Please find a detailed response to the reviewer’s comments in the attached file.  

Reviewer 2 Report

Comments and Suggestions for Authors

firstly, I’d like to thank you for the opportunity to review this manuscript. The research problem presented in the submitted paper is interesting and worth to be studied. The text is well-referenced, tables (3) and a figure (1) are easy to understand. My suggestions:

1. Keywords: try not to repeat the same words from the title.

2. Introduction. This section is really short. On the other hand, you have a Literature review, which is well-written.

3. Methods. You used data from the 2006 HRS survey. Don’t you have updated data? Or, are there any limitations to the usage of the new ones?  In this case, you present the state that we had 17 years before. It makes your study not up-to-date, but more a historical insight. I think it should be mentioned in the Introduction and in the Abstract. This section could be clarified. Moreover, it would be more reader-friendly with a research flowchart.

4. Discussion, in my opinion, is too short.

Kind regards.

Comments on the Quality of English Language

Minor editing of English language required.

Author Response

We appreciate the constructive comments provided by Reviewer 2. We are confident that the manuscript has been significantly improved following the revisions made in response to the reviewer’s valuable suggestions. Please find a detailed response to the reviewer’s comments in the attached file.  

Reviewer 3 Report

Comments and Suggestions for Authors

This is an interesting and well-written article.  I especially appreciate the data display in Figure 1. I do have a few comments that might help to maximize the scientific contribution of the paper. 

1. While the covariates included in the analysis help to strengthen the implicit causal argument, it would be very helpful if the authors acknowledged the inherent limitation in using self-report time-related data embedded in a cross-sectional panel.  At a minimum, the relationship between travel and the outcome variables is likely to be reciprocal; at maximum, the causal direction could well be from the variables described as dependent in this paper (cognitive function, loneliness, and depressive symptoms) to the designated IV of travel. 

2.  The way that distance of travel is described in the discussion is excellent (e.g., that distance is a proxy for complexity and required engagement). I'd like to see that conceptual framing embedded in the "measurements" section also.  

3.  There are two places in the article that suggest implications of the finding for therapeutic interventions (page 9) and business/program/service development  (page 3).  That may be a reasonable assumption, but there is a lot territory between the findings of this study and any kind of new program or intervention.  Can you help readers understand what that path might look like?  If not, I'd drop these claims.

Author Response

We appreciate the constructive comments provided by Reviewer 3. We are confident that the manuscript has been significantly improved following the revisions made in response to the reviewer’s valuable suggestions. Please find a detailed response to the reviewer’s comments in the attached file.  
